# Towards Human Motion Tracking Enhanced by Semi-Continuous Ultrasonic Time-of-Flight Measurements

**DOI:** 10.3390/s21072259

**Published:** 2021-03-24

**Authors:** Silje Ekroll Jahren, Niels Aakvaag, Frode Strisland, Andreas Vogl, Alessandro Liberale, Anders E. Liverud

**Affiliations:** 1SINTEF Digital, 0373 Oslo, Norway; esilje@gmail.com (S.E.J.); niels.aakvaag@sintef.no (N.A.); frode.strisland@sintef.no (F.S.); andreas.vogl@sintef.no (A.V.); alessandro.liberale@sintef.no (A.L.); 2Department of Physics, University of Oslo, 0371 Oslo, Norway

**Keywords:** human motion tracking, gait analysis, inertial measurement units, body sensor network, ultrasonic time of flight

## Abstract

Human motion analysis is a valuable tool for assessing disease progression in persons with conditions such as multiple sclerosis or Parkinson’s disease. Human motion tracking is also used extensively for sporting technique and performance analysis as well as for work life ergonomics evaluations. Wearable inertial sensors (e.g., accelerometers, gyroscopes and/or magnetometers) are frequently employed because they are easy to mount and can be used in real life, out-of-the-lab-settings, as opposed to video-based lab setups. These distributed sensors cannot, however, measure relative distances between sensors, and are also cumbersome when it comes to calibration and drift compensation. In this study, we tested an ultrasonic time-of-flight sensor for measuring relative limb-to-limb distance, and we developed a combined inertial sensor and ultrasonic time-of-flight wearable measurement system. The aim was to investigate if ultrasonic time-of-flight sensors can supplement inertial sensor-based motion tracking by providing relative distances between inertial sensor modules. We found that the ultrasonic time-of-flight measurements reflected expected walking motion patterns. The stride length estimates derived from ultrasonic time-of-flight measurements corresponded well with estimates from validated inertial sensors, indicating that the inclusion of ultrasonic time-of-flight measurements could be a feasible approach for improving inertial sensor-only systems. Our prototype was able to measure both inertial and time-of-flight measurements simultaneously and continuously, but more work is necessary to merge the complementary approaches to provide more accurate and more detailed human motion tracking.

## 1. Introduction

Human motion tracking is a systematic tracking and analysis of quantitative or qualitative features of human motion, such as movement classification, evaluation of physiological and biomechanical parameters of motion, gait function, work ergonomics or performance/technique in sports [1]. Human gait consists of cyclic movements of the limbs, which are repeated for each stride. Gait abnormalities can indicate disease, and progression in neurological diseases can often be tracked by accompanying gait parameter changes. Gait analysis is therefore employed both for therapeutic and diagnostic purposes. For example, the timed 25-foot walk test (T25FW) is a standard diagnostic lower limb test performed on persons with multiple sclerosis (pwMS) [2,3]. Quantitative assessment of gait parameters can aid clinicians by providing supporting evidence, predicting fall risk and quantifying the effect of treatments. Many gait parameters are, however, difficult to assess by observation alone due to their short duration. In top-level sports, motion tracking is even more focused on intracycle performance and details, and accurate movement observations are needed to adjust the technique to maximize performance for the individual athlete. An experienced coach can observe very subtle details in the movement patterns of his or her athletes and use this as basis for subjective feedback. However, without a motion tracking system, these observations cannot be stored and used in systematic and objective analysis independently of the coach.

Camera-based 3D motion capture systems with body markers, such as Vicon (Vicon Motion Systems Ltd., Oxford, UK), Qualisys (Qualisys AB, Gothenburg, Sweden) or Optotrak (Norther Digital Inc, Ontario, Canada), are considered the gold standard in terms of accuracy in motion tracking and gait analysis [4]. However, these are rather expensive systems, and the fixed camera set-ups limit them predominantly to lab usage [1,5,6]. Kinect (Microsoft Kinect, Microsoft Corporation, Washington, DC, USA) or other RGB depth cameras can be used as less expensive alternatives to measure distance and velocity aspects of motion, but most joint angles are difficult to measure with these cameras [6].

Wearable motion tracking systems mainly employ Inertial Measurement Units (IMUs) as sensors. IMUs normally consist of three axis accelerometers, often complemented by three axis gyroscopes and/or three axis magnetometers, and are now widely available as miniaturized electronic components [1,4,5,7]. IMUs attached to the human body can measure motion data at the specific sensor attachment locations, and multiple IMUs enhanced with wireless communication can gather real-time synchronized data in a wireless Body Sensor Network (BSN) [4] that can be used continuously in out-of-the-lab environments [5]. The main disadvantages of IMUs are their limitations in accuracy due to errors, bias offsets, and zero-point drift over time. In particular, the estimation of positions (double integration of the acceleration) and angles (integration of angular velocity) leads to errors and drift over time of even very accurate sensors unless special recalibration precautions are implemented, such as intracycle or other a priori-based recalibration approaches. The required timing synchronization between sensors is an additional complication for BSNs. Furthermore, IMU BSNs that have sensor units distributed over the body are unable to keep track of the relative positions of the individual IMUs, and must therefore rely on a priori assumptions in order to analyze the data correctly. For example, the data analysis system will have to know the attachment location of all sensor devices, and must assume that sensors do not shift location during the activity. For a system with IMUs located on the feet, an accurate methodology combining zero-velocity recalibration and Kalman filtering has been established and well-tested [4,8]. Teufl et al. (2018) [9] showed that IMU-based gait analysis systems have relatively high errors in spatial parameters (RMS error 6–7% step length, 35% for step width and swing width), but seem promising for other parameters (RMSE 0.9–4.4%). Mariani et al. (2010) [10] described and validated a system using the S-sense IMU (which later became Physilog^®^ (Gait Up SA, Renens, Switzerland)) for assessment of 3D gait spatial parameters in young and elderly subjects during straight walking and turning. They achieved an RMS error of 6.5% in step length. Gait analysis using two foot-worn Physilog^®^ 5 IMUs has also been validated with high reliability for individuals affected by stroke [11,12]. In spite of the progress made, a complete wearable sensor motion tracking system suitable for usage by nonexperts outside controlled laboratory settings is still not available [4], due largely to the lack of robust approaches to automated BSN calibration, synchronization and sensing of relative positions.

Ultrasonic time-of-flight (ToF) measurements can be used to measure relative distances. Zizzo and Ren (2017) [13] found a 15% reduction in total accumulated position and step length errors during indoor walking in a combined IMU and ultrasonic ToF system, relative to IMUs alone. Qi et al. (2016) [14] demonstrated the feasibility of a wearable ultrasound-only gait monitoring system for long term monitoring at home. Ashhar et al. (2017) [15] used a system with ultrasonic sensors on the legs to measure distances to an array of wall mounted ultrasonic transducers to analyze gait. ToF measurements combined with IMUs have also been employed for indoor navigation and showed improved accuracy when compared with IMU-only systems [16,17,18].

Most systems using ToF measurements for human motion tracking use ultrasound frequencies of 40 kHz [13,14,15,16,17], which correspond to wavelengths in air of around 1 cm. It would thus be desirable to increase the ultrasound frequency to improve accuracy and to enable semicontinuous intracycle measurements.

The aim of the current study is to investigate the feasibility of using a high frequency (70–110 kHz) ultrasonic ToF sensor to enable semicontinuous measurements of relative limb-to-limb distance, and to build a prototype including both IMU and ToF sensors. In this study, we focus on testing a ToF sensor in dynamic gait tests and compare it with a commercial IMU. This is part of a larger effort to investigate if the relative dynamic distance between the IMUs can be used to improve the relative position estimation and angle estimation compared to using IMUs alone.

Following this introduction, we will introduce the design goals for the IMU and ToF motion tracking system in Section 2, followed by a description of the realized prototype and characterization of the system. Finally, a description is given of the gait test protocols using the prototype in the lab and outdoors. Section 3 presents results from the gait tests, followed by a discussion of the results in relation to system objectives in Section 4.

## 2. Materials and Methods

### 2.1. Human Motion Tracking Technology Needs

As design input, and to ensure relevance and usefulness of sensor configurations, we carried out end-user phone interviews with six human motion tracking experts (one exercise physiologist, two physiotherapists, one neurologist and two physicists), all regularly involved in human motion analysis. The end-user interviews were approved by the Norwegian Center for Research Data (NSD: https://www.nsd.no/en, 23 March 2021). The participants gave oral and written consent prior to participation and could withdraw from the study at any time. It is not the objective of this study to meet all their stated requirements, but rather to use these as guidance for design choices made in the current explorative test and feasibility evaluation.

The experts’ current set of methods include visual observations, pictures, video recordings, stopwatch, IMUs, paper forms, standardized physical tests, neurological tests, and multisensory systems in combination with video recording. For new systems, a general request from all of them was for more objective measures of motion, such as tracking of the center of mass, relative measure between limbs and measurements of limb/joint angles. Hip function and movements were considered difficult to assess, and the many joints and bones make hand function analysis complex. It was further emphasized that classification of balance, force and technique was more challenging than assessments of speed, acceleration, and frequencies. Multiple distributed sensor units are considered demanding to use for several reasons, including mounting time, starting and stopping procedures, as well as all the practicalities associated with use, data collection, time synchronization between devices, calibration, configuration, and analysis. Systems that might more easily enable comparison of measurements taken from different users or at different times were requested. Finally, the experts desired systems for out-of-lab or in-the-field measurements, preferably supporting long term measurements of up to several days continuously.

When it comes to physical characteristics, sensor units should be comfortable, not interfere with motion, be small in size, unobtrusive and lightweight—e.g., comparable with current state-of-the-art devices. The devices should be designed to be simple to use, preferably with automatic calibration procedures, automatic establishment of communication and time synchronization between different sensor nodes (e.g., centralized simultaneous start and stop of all units). Device accuracy should be comparable or better than current IMU systems, which have a gait distance accuracy in the range of (a couple of) centimeters. Devices should be easy to attach, be clearly marked, and the design should aim to keep the required number of sensor devices as low as possible. In addition to hardware aspects, excellent data feedback and analysis software, preferably offering real-time feedback, is valued since it can enable real-time input on system operation, identify system failures, and provide direct feedback to the sensor device user or a system observer. It is also desirable to enable data annotation in real time, in some cases also combined with video recording, to support quick retrospective data analysis of sequences of particular interest. Finally, the system should be able to produce autogenerated case-specific reports after a measurement sequence to allow compact summaries for export to, e.g., a patient journal or training logbook.

### 2.2. Hardware System Description

The developed prototype system consists of sensor module pairs where each has one integrated 3D accelerometer, 3D gyroscope, 3D magnetometer IMU with digital readout (LSM9DS1, STMicroelectronics, Geneva, Switzerland), and either an ultrasonic loudspeaker (custom-made, SINTEF, Oslo, Norway) or an ultrasonic microphone (Knowles, Itasca, IL, USA). Figure 1 shows the module concept and building blocks. The module pairs can provide inertial sensor data (3D acceleration and angular velocity) of each module, as well as the relative ToF distance between the modules. The modules are integrated on a development board (Nordic Semiconductor, Trondheim, Norway) with a microcontroller and a radio frequency wireless communication unit, a custom-made printed circuit board for ultrasound and IMU electronics as well as a battery. The modules are controlled from a central node, which also receives the measured data from the ToF sensor and the IMUs.

#### 2.2.1. Ultrasound Technology

The ultrasonic loudspeakers in the system are piezoelectric micromachined ultrasonic transducer (PMUT)-based ultrasonic transducers developed at SINTEF. These ultrasonic loudspeakers are based on SINTEF’s thin-film PZT (lead zirconate titanate, 2 µm film thickness) bulk micromachining microelectromechanical system (MEMS) technology. The thin-film PZTs have been deposited by high quality chemical solution deposition. The piezoelectric coefficient e_31,f_ of these films is −18 C/m^2^. The transducers are based on a membrane structure made up of 8 µm silicon-on-insulator, 1 µm stress compensation oxide, a platinum bottom electrode, 2 µm PZT, and a gold top electrode. The MEMS loudspeaker is 1.7 × 1.7 × 0.4 mm in size. The loudspeaker devices were characterized with respect to frequency response of the sound pressure level for different actuation voltages:95–100 dB @ 10 cm;sound pressure at resonance saturated already at 5 V AC;bandwidth continues to widen using higher voltages (−6 dB BW ~14% using 10 V AC).

The ultrasonic microphone in the system is a commercial Knowles analogue silicon MEMS microphone, SPH0642HT5H-1, with an acoustic sensor, a low noise input buffer and an output amplifier. The microphone is stated to have a flat frequency response from 100 Hz to 10 kHz. We tested that the microphone operates out of range in the ultrasound frequency range. The SINTEF PMUT loudspeaker was used to send a frequency sweep from 70 to 110 kHz, and around the resonance frequency of the loudspeaker (90–100 kHz) the receiver microphone signal was found to have sufficient quality to be used in the present experiments. Figure 2 shows the frequency response of the microphone resulting from the frequency sweep transmitted by the SINTEF PMUT loudspeaker.

#### 2.2.2. Ultrasound and IMU Electronics

The purpose of the electronic circuit board is to generate suitable excitation signals for the loudspeaker and to amplify the signals received from the microphone, where the noise level should be as low as possible. The transmitter and receiver boards are identical, except that the transmitter board was equipped with a loudspeaker while the receiver board was fitted with a microphone. The boards also contain an inertial measurement module. The electronic circuit board has the following main components (Figure 3): a 3V Low Dropout (LDO) regulator, a 12V charge pump DC/DC converter, an LSM9DS1 inertial measurement unit (IMU), a SINTEF PMUT loudspeaker (TX), a Knowles SPH0642HT5H-1 microphone (RX), an astable multivibrator pulse (AST) generator and an amplifier circuit, as well as several resistances and capacitors. The power supply is from a Nordic Semiconductor developer board (see Section 2.2.3) where a 5V DC supply is available. A custom-made dedicated circuit was used to produce a 12V loudspeaker excitation signal. Another custom circuit was used to amplify the microphone signal in front of the microcontroller ADC.

#### 2.2.3. Microcontroller and Development Board for the Ultrasonic ToF Sensor

An nRF52 development kit from Nordic Semiconductor was used as a basis module as it contains a radio with an embedded microcontroller. The software may be configured to accommodate different radio protocols such as Bluetooth Low Energy, Thread, ANT, and Enhanced Shock Burst (ESB). The kit also allows easy access to some basic peripherals such as GPIOs and a 14-bit analog-digital converter (ADC). Powering of the board and the other electronic components was carried out using a power bank (MO8735, Mobility on board (MOB), Paris, France). Figure 4 shows a block-schematic representation of the hardware setup.

The distance between the sensors was estimated using a ToF measurement of the ultrasonic pulse. Accurate range estimates require a common timing reference for both the loudspeaker and microphone modules. A distance accuracy of 1 cm dictates a timing accuracy of approximately 30 µs. This is achieved by the loudspeaker modules sending a short radio packet prior to emitting the ultrasonic pulse. Timing uncertainties can arise from delays in the protocol stack, both for the loudspeaker and microphone. To minimize the uncertainty, a simple real-time protocol (ESB) was chosen where both listen-before-talk (LBT) and retransmissions of lost packets may be switched off, as both these mechanisms introduce uncontrolled delays. Similar synchronization solutions using low latency wireless systems have also been reported in [19,20]. The remaining stack delays were all constant values and could easily be adjusted in the software.

The loudspeaker sent bursts of 20 cycles of 100 kHz ultrasonic pulses. On the receiving side, the electronics filter and amplify the signal and present it to the ADC with a 1.65 V DC bias. This signal was sampled at 150 kHz and digitally processed to provide an estimate of the elapsed time. The maximum sampling frequency provided by the nRF52 was 200 kHz, which is at the Nyquist rate. It was therefore decided to subsample to avoid sampling too close to the signal DC crossing.

The presence of a signal will cause an oscillation around the abovementioned 1.65 V DC bias, where the sign of the deviation is irrelevant. Hence, only the absolute value of the signal needs to be considered. The algorithm employed for the energy detection can thus be expressed by the following relation:(1)e(n)=F{abs(x(n)−1.65)}
where x(n) is the signal presented to the ADC, 1.65 is the DC bias, and *F*{} is a moving average filter. The ToF was estimated to where the signal e(n) crossed a certain threshold that was experimentally determined.

### 2.3. Characterization of the ToF Sensor

The lobe, width and length of the ToF sensor were tested in two different static tests (Figure 5). In a first test, the loudspeaker module was kept at a fixed position, and the microphone module was placed at 10 to 100 cm distance (L) to the loudspeaker in steps of 10 cm. In the second test, the microphone was kept fixed at L=30 cm distance to the loudspeaker, and the angle of the loudspeaker ($\Theta_{L})$, to the central line between the two ultrasonic transducers, was changed from 0° to 90° in steps of 10°. Static ToF distance was measured continuously at 10 Hz (10 distance samples per second) for a minimum of 10 s at each position during the two tests.

The ToF sensor lobe had a range of up to 60 cm at the current measurement configuration. The lobe width was wide with sufficiently strong signals for ToF measurements from 0° to 90° between the loudspeaker and the microphone. Figure 6 shows the measured distance for the lobe range (a) and the lobe width (b). The root-mean-square errors (RMSEs) for the different distance measurements in (a) were in the range 2.3–3.3 mm, except for the 600 mm distance measurement which had an RMSE of 6 mm. The average RMSE was 3.4 mm. For the different angles (b), the RMSE for the distance measurements ranged from 1.8 to 5.2 mm with an average of 3.1 mm.

### 2.4. Dynamic Gait Test Protocols

Two dynamic gait test protocols (treadmill walking and outdoor walking) were performed with two prototype modules, one loudspeaker module and one microphone module attached to the lower legs of two test subjects. The aim was to carry out semicontinuous measurements with a beam pattern being transmitted from the transducer to the microphone at the lower legs. The time of flight was measured when the signal was received and was converted to a distance measurement. The measurement was repeated providing a semicontinuous measurement of the distance between the legs in the stride cycle. The measured stride lengths were compared with the stride lengths from the validated commercial GaitUp ^®^ IMU sensors [11,12,21] attached to the feet of the test subjects, as well as the calculated length from actual walked distance divided by number of steps. The prototype modules should ideally be attached to the feet together with the GaitUp ^®^ IMU sensors; however, due to the size of the prototype and the range limit of 60 cm, the sensor modules were attached at the lower leg. The integrated IMUs in the prototype modules were not used in these dynamic test protocols. Video recordings of the test subjects during the gait tests were used as ground truth. Figure 7 shows the sensor setup during the two dynamic gait tests. The ToF measurements were recorded at 10 Hz, the GaitUp ^®^ IMUs at 128 Hz, and the video recordings at 30 frames/seconds. The test subjects carried the central node to maintain the Bluetooth connection.

The current study did not aim to produce new knowledge within medicine or other health topics and was therefore not subject to evaluation and/or approval by a medical research ethics committee under the applicable regulations. The two study participants, who are also coauthors of this paper, consented to take part in the experiments. They had full insight into study aims, benefits and potential risks—the latter was evaluated by the whole project team to be very low—and volunteered to take part as participants and agreed that data from the experiments may be published.

#### 2.4.1. Treadmill Walking

One pair of modules was tested during walking on a treadmill in a lab environment. The ToF sensors were attached to the lower legs, with the loudspeaker module attached to the right lower leg and the microphone module to the left lower leg. Two GaitUp ^®^ IMU sensors were attached to the left and right foot. A 5-min walking test was performed by two different test persons on the treadmill at slow speed (0.8 km/h). A slow speed was selected to keep the distance between the ToF sensors small and thereby stay within the range of the sensors. Treadmill walking speed was measured using the length of the band along with the revolution time for the band, as observed in the recorded video.

#### 2.4.2. Outdoor Walking

One pair of modules was tested in a dynamic gait test in an outdoor (out-of-the-lab) environment. The ToF sensors were attached to the lower legs, with the loudspeaker module attached to the left lower leg and the microphone module to the right lower leg. Two GaitUp ^®^ IMU sensors were attached to the left and right feet. A 25-foot walking test was performed by two different test persons using the following protocol:Stand still at the first mark for 5–10 s.Walk 25 feet towards and past the second mark.Turn and go back to the second mark.Stand still at the second mark for 5–10 s.Walk 25 feet towards and past the first mark.

### 2.5. Data Analysis

The data from the different sensor units (the ToF sensors and the GaitUp ^®^ IMUs) were synchronized with the video recordings using a custom-made software (AutoActive research environment: https://www.sintef.no/projectweb/autoactive/, 23 March 2021) and imported in MATLAB ^®^ (The Matworks, Inc., Natick, MA, USA) for further analysis. Stride lengths were calculated using the validated GaitUp ^®^ LAB software [11,12,21] from the GaitUp ^®^ IMUs on the feet.

An average stride length was additionally calculated from the actual walked distance divided by the number of steps. The ToF sensors attached at the lower legs have different pendulum lengths compared to the GaitUp ^®^ IMUs attached to the feet (see sensor positions in Figure 7), and therefore they measured different lengths. To compare the measured relative ToF distances to the stride lengths measured using the GaitUp ^®^ IMUs and the actual walked distance, we recalculated the stride lengths between the feet to estimate the stride lengths at the height of the ToF sensors. This was carried out by measuring the lengths of the pendulum at both positions and assuming straight legs when the feet are maximally separated.

## 3. Results

Figure 8 shows an example of the measured semicontinuous (10 measurements per second) ToF distance between the lower legs of a test person during a 25-foot walk. The ToF distance oscillated between a short distance, when the two legs crossed each other, and a larger distance, when the two feet were far away from each other. There are only about eight ToF distance data points per cycle (minimum to maximum and back to minimum ToF distance), and the peaks are clearly uneven and differ in height, indicating that we do not have enough measurement points to measure the real peak values due to the low sampling frequency (10 Hz). The same can be observed for the minimum values. As indicated in Figure 8, some data points are lost. During the 25-foot walk tests outdoors, about 5% of the data points were lost, while during the treadmill tests about 35–55% of the data points were lost. The large data loss at the treadmill could have been caused by increased noise indoors and more objects close to the test subjects but may also have just been due to instability of the protype or bad connection with the central node. We believe, however, that most data points were lost when the feet were close together during the strides and the relative distance was well below 20 cm, and thereby out of range for the ToF sensor.

### 3.1. Treadmill Walking

Figure 9 shows the distribution of measured maximum ToF distances and estimated GaitUp ^®^ IMU stride lengths (for both left and right leg) at the height of the lower legs (see ToF sensor positions indicated in Figure 7) of test persons 1 and 2. For test person 1, the maximum ToF distance per stride between the lower legs ranged from 139 to 563 mm (average: 235 mm, SD: 54 mm) during the walking cycle, whereas the corresponding GaitUp ^®^ IMU estimated stride lengths ranged from 182 to 314 mm (average: 242 mm, SD: 23 mm). For test person 2, the maximum ToF distance per stride between the lower legs ranged from 141 to 357 mm (average: 236 mm, SD: 44 mm) during the walking cycle, whereas the corresponding GaitUp ^®^ IMU estimated stride lengths ranged from 110 to 323 mm (average: 247 mm, SD: 20 mm).

The actual walked distance for both test subjects was 138.8 m at a speed of 0.46 m/s (measured from treadmill band length and number of band cycles, both walked 5 min). The number of strides for test subject 1 was 369, with an average stride length of 376 mm between the feet and an estimated average of 229 mm between the lower legs. The number of strides for test subject 2 was 354, with an average stride length of 392 mm between the feet and an estimated average of 258 mm between the lower legs. The number of strides were equal for all the different sensor measurements and the video records for both test subjects.

The measured ToF distances were on average shorter than the estimated GaitUp ^®^ IMU stride lengths, and had a higher SD. Additionally, the ToF distances had a larger spread compared to the estimated GaitUp ^®^ IMU stride lengths. However, the range of ToF distances were found to overlap with the range of estimated GaitUp ^®^ IMU stride lengths as well as the average stride length (blue boxes in Figure 9).

### 3.2. Outdoor Walking

Figure 10 shows the distribution of measured maximum ToF distance and estimated GaitUp ^®^ IMU stride lengths (for left and right legs) between the lower legs for both test subjects. For test subject 1, the ToF distance ranged from 239 to 498 mm (average: 380 mm, SD: 46 mm) with GaitUp ^®^ IMU estimated stride lengths ranging from 335 to 446 mm (average: 384 mm, SD: 31 mm). For test subject 2, the ToF distance ranged from 246 to 480 mm (average: 342 mm, SD: 40 mm) with GaitUp ^®^ IMU estimated stride lengths ranging from 186 to 376 mm (average: 330, SD: 45 mm).

The measured maximum ToF distances were on average shorter than the estimated GaitUp ^®^ IMU stride lengths and had a larger SD. As for the treadmill test, the maximum ToF distances had a larger spread compared to the estimated GaitUp ^®^ IMU stride lengths. However, the range of maximum ToF distances were found to overlap with the range of estimated GaitUp ^®^ IMU stride lengths as well as the average stride length (blue boxes in Figure 10).

## 4. Discussion

In this study, we have investigated the feasibility of using high frequency (70–110 kHz) ultrasonic ToF sensors to enable relative limb-to-limb distance measurements, and the feasibility of building a sensor prototype that integrates both IMU and ToF sensors. We believe that ToF measurements of relative distance between the IMUs can be used to improve the relative position estimation and angle estimation compared to using IMUs alone. If this can be achieved, it might enable a system of IMUs that is easier to configure and use, and that yields more accurate and robust measurements of gait parameters in both in-lab and out-of-the-lab environments. In this study, the system has been tested for human gait both in a controlled lab environment on a treadmill, as well as an outdoor walk test. The system operates as intended and enables semicontinuous measurement of relative distances between the sensor modules, and with step length estimates comparable to commercial IMUs. The ultrasonic ToF measurements are closer to the average stride length, but with a larger standard deviation compared to the IMUs (Figure 10). This larger standard deviation is expected due to the low sampling frequency (10 Hz) of the ToF measurements, which under-sample the dynamic gait motion. With a higher sampling frequency, this problem would be reduced, and it is expected that the combination of ToF and IMU measurements would even improve the accuracy further. Other studies have shown that ToF measurements used together with IMUs for indoor navigation can improve accuracy compared to IMU-based systems alone [16,17,18].

The ToF sensor was able to capture the dynamic pendulum motion of the lower legs (Figure 8) and the relative distance between them. The measurements show, as expected, that the distance was largest when both feet touched the ground during a stride and smallest when the legs passed each other in the middle of the stride with one foot in the air and one on the ground. However, due to occasional data package losses of the prototype or some missed ultrasonic pulses, some measurement points were lost. The occasional package losses are thought to have little impact on the performance as the controller would maintain sufficient synchronization due to the slow drift in the oscillator. The missed detection of ultrasonic pulses (5% loss was seen outdoor, and 35–55% indoor at the treadmill), however, causes more problems. Increasing the transmit power and/or selecting units with a wider lobe can improve the accuracy. An additional issue is that the ultrasonic transmission path tends to be obstructed by the leg at maximum stride. This is a fundamental problem caused by the placement of the units and can be overcome only by a total redesign of either the loudspeaker or the microphone. For example, replacing the transducer with one on the front side and one on the back side of the leg, will ensure 360° coverage with no path obstruction.

The current prototype module pair measured stride lengths (maximum relative distance between the legs during gait) that was comparable to the stride length estimated by the commercial GaitUp ^®^ IMUs (Figure 8 and Figure 10). Both the prototype and the commercial GaitUp ^®^ IMUs measured a mean stride length comparable with the average stride length measured by the actual walked distance divided by number of strides. The stride lengths were all in the same range; however, the prototype showed larger variations and bigger standard deviations. The ultrasonic pulse repetition of the ToF sensor in the current prototype is only 10 Hz. This gave the ultrasonic pulses time to die out between transmissions to avoid erroneous detections. However, it did limit the time resolution of the received position data. It is therefore expected that the prototype is not able to accurately detect the peak values of the relative distance (maximum and minimum distance between the legs), due to the continuous motion of the legs and the walking frequency (>1 per second) which will acquire a higher sampling rate. This can explain the larger variations of the stride length compared to the GaitUp ^®^ IMUs despite the more accurate measurement method of relative distance. In a future implementation, the pulse repetition frequency should be increased to preferably above 30 Hz, which would better capture the details without compromising reliability and still allow the ultrasonic pulses to die out before the next pulse.

The ultrasonic loudspeaker transmitted a series of regular pulses at a fixed frequency (100 kHz), and the received microphone signal was sampled at 150 kHz. This is below the Nyquist limit and is therefore a suboptimal solution. Increasing the sampling rate would have resulted in a finer timing resolution. However, the ADC of the current system had a limit of 200 kHz. Sampling narrow band signals at the Nyquist rate caused sporadic signal loss due to the occasional sampling at signal zero crossings. A future implementation should be designed to support higher sampling frequency. Compared to similar studies of gait using ToF measurements, the ultrasonic pulse frequency of the current study was higher (100 kHz instead of 40 kHz) [13,14,15,16,17], which is expected to possibly increase accuracy and time resolution of the ToF. Currently, the estimated accuracy of 1 cm the threshold-based ToF detection system was considered sufficient. Another option for the ToF system could be to transmit chirps (a short sweep signal over a given frequency range) instead of constant frequency pulses. A chirp-based system could provide better accuracy for the time of arrival for the signal by matching frequency at the receiver side. However, generation of chirps requires more complex transmitter electronics, and the receiver must carry out more processing to detect the time of the received signal. The human motion tracking system is meant to be portable and battery operated, hence power usage should be kept low to ensure sufficient operational time. Application of the Doppler effect can be used for measuring speed of the limbs in continuous motion. In this case, the apparent frequency change caused by the relative velocities between the loudspeaker unit and the microphone can be used to calculate velocity. The speed of limb motion can also be measured using the distance between successive measurements. The experiments have shown that the current prototype does not provide data with sufficient accuracy and resolution for useful velocity measurements. However, there are not expected to be any further barriers to achieving this independent relative velocity measurement, beyond further optimizing the sensor configuration in terms of ultrasonic sampling frequency and the design of the ultrasonic pulse profile.

The semicontinuous ToF distance measurement can be useful for aligning and constraining movements of sensor pairs relative to each other. Previous studies of, e.g., stride length using ToF measurements have been noncontinuous measurements during the double support phase of the gait [13,16]. A semicontinuous distance measurement of dynamic movement could provide additional information about the full movement pattern instead of single parameters. However, the ToF sensor has the drawback that it needs a line of sight between the modules. This makes it difficult to measure distances between, for example, the left leg and right arm since the user’s body will tend to cover the free line of sight between the modules. In effect, this could possibly lead to overestimation of distances because the ultrasonic pulse pattern must propagate around the obstructing body, or missing measurements because well-defined pulse patterns cannot be detected. Furthermore, the modules must always be within the lobe of the other modules to be able to communicate. In the current study, we have only tested the feasibility of a single module pair between the lower legs of the test subjects to measure relative distance during gait.

Extending the approach of ultrasonic ToF relative distance measurements even further, it could pave the way to realizing a human body coordinate system involving multiple sensors distributed in a body sensor network. Presently, it is practically impossible to analyze IMU data from multiple body-worn sensors unless additional a priori assumptions are made—for example, that body limbs have fixed lengths and certain movement patterns. However, extending the system to multiple devices could quickly introduce ultrasonic bandwidth problems, so this might be required to use multiple ultrasonic pulse frequencies to identify and keep track of the various loudspeakers and when the respective ultrasonic pulse train arrives at the various, distributed microphones. One additional challenge with referencing more than two units is linked to mirror symmetries. In short, two sensors detecting a third sensor cannot determine the direction to the third sensor—only that it lies in a circular orbit with respect to the two “reference” sensors. Although mirror symmetries represent a significant complication of the measurement problem, these problems can probably, in many cases, be overcome by combining the ToF measurements with the corresponding IMU data. The establishment of a human coordinate system has not been tested as part of this study but represents an interesting motivation and way forward for ultrasonic ToF + IMU sensor systems. However, we believe that such a system could improve the analysis of human motion by adding more detailed and objective measures of motion. Especially, detailed intercycle information and quality of motion could be helpful to aid clinicians to improve diagnosis and therapy in neurological diseases with motion impairments, or help top athletes and coaches to track and improve sport performance.

The current prototype has several limitations. The sampling frequency of 10 Hz is too low to achieve sufficient accuracy in the semicontinuous distance measurements during gait. The system design is also limited by the ultrasonic transducer lobe and field of view causing loss of ultrasonic pulses and thereby data points. Integrating several transducers and microphones in each module could mitigate part of this problem. The design is also rather bulky and needs to be miniaturized for a larger system with sensor modules on multiple limbs. The sampling rate of the received ultrasonic signal at the microphone is limited by the speed of the controller board, resulting in inaccuracy in the ToF measurements.

Based on our experience designing and using our prototype, we consider the integration of an ultrasonic loudspeaker and microphone into a commercial IMU-based sensor as feasible. The ultrasonic loudspeaker, microphones and control electronics are all low power components; hence, the components can be integrated into a battery powered IMU + ToF device without significantly reducing operational time between charging the unit. The required sizes of the loudspeaker, microphone and electronics can all be made very compact down to 2 × 2 mm^2^ each and should not have a significant impact on the overall size of the sensor module. However, the location and shielding of the loudspeaker and microphone must be carefully selected to provide repeatable, valid distance measurements between sensor devices and ruggedized to enable use in realistic conditions and operational environments.

## 5. Conclusions

In this study, we demonstrate a prototype providing ToF distance data in addition to IMU data for human motion tracking. The recorded data show that this is feasible for human gait measurements, but improvements in sampling rate and strategies in cases where the line of sight between sensors is obstructed will be needed. An enhanced system with a higher sampling frequency and a wider angle of operation could provide valuable distance and speed data between limbs and could be used to improve the relative position estimation and angle estimation compared to using IMUs alone. Such a system could be used as an out-of-the-lab gait human motion tracking system.

## Figures and Tables

**Figure 1 sensors-21-02259-f001:**
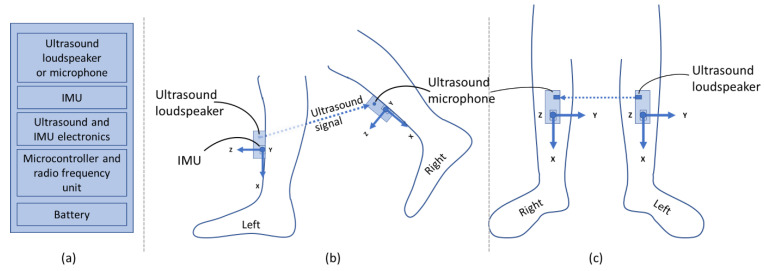
System concept (**a**) with one pair of modules attached to the lower leg seen from the side (**b**) and from the front (**c**). The modules consist of an Inertial Measurement Unit (IMU) and either an ultrasonic loudspeaker (in this case, left leg) or a microphone (in this case, right leg). The ultrasonic signal is marked with a dashed arrow, and the coordinate system of the IMUs with solid arrows.

**Figure 2 sensors-21-02259-f002:**
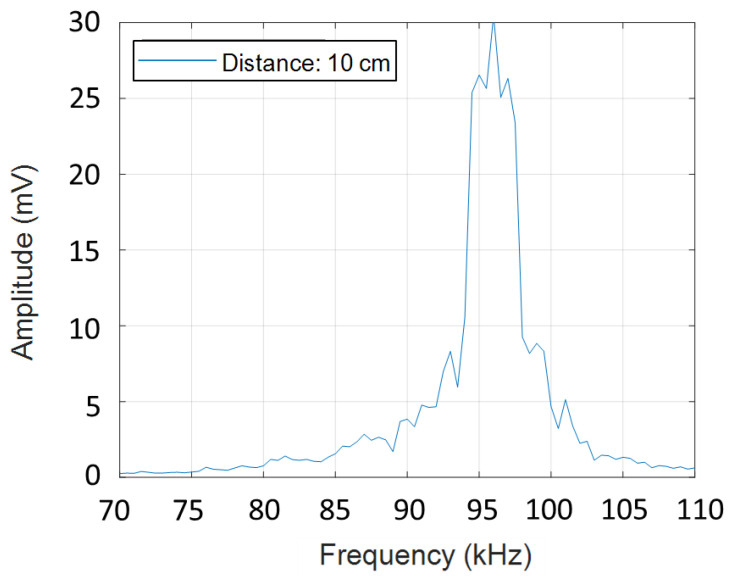
Frequency response of the received raw signal of the Knowles microphone transmitted by the SINTEF piezoelectric micromachined ultrasonic transducer (PMUT) loudspeaker at 10 cm distance.

**Figure 3 sensors-21-02259-f003:**
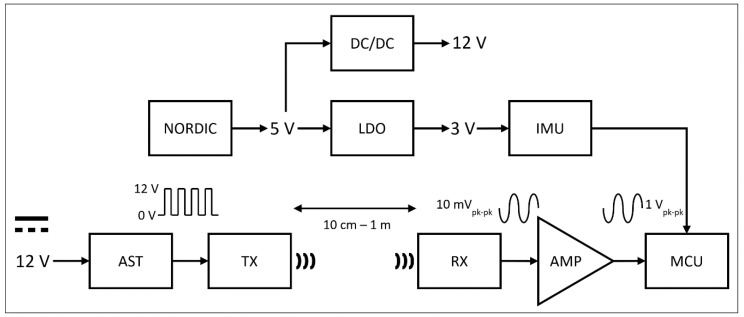
Transmitter and receiver electronics boards functional building blocks supported by text description: NORDIC: Nordic Semiconductor development board. DC/DC: charge pump. LDO: low dropout regulator. IMU: inertial measurement unit. AST: astable multivibrator pulse generator. TX: ultrasonic loudspeaker (transmitter board only). RX: ultrasonic receiver (microphone board only). AMP: amplifier circuit. MCU: microcontroller.

**Figure 4 sensors-21-02259-f004:**
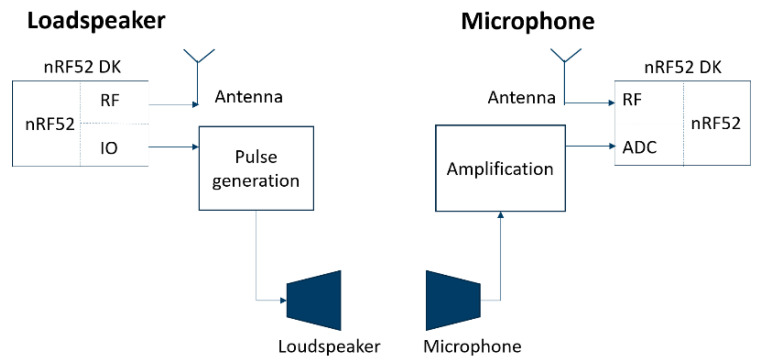
Block schematic representation of the hardware setup of the sensor module pair.

**Figure 5 sensors-21-02259-f005:**
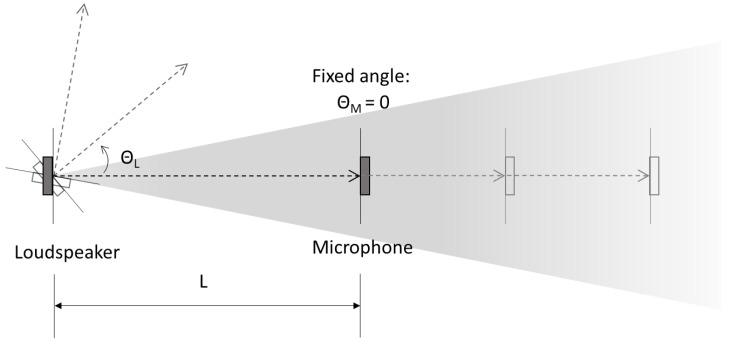
Illustration of the time-of-flight (ToF) sensor characterization setup. L: The distance between the loudspeaker and the microphone. $\Theta_{L}$: loudspeaker angle to the central line between the loudspeaker and the microphone. $\Theta_{M}$: microphone angle to the central line between the loudspeaker and the microphone.

**Figure 6 sensors-21-02259-f006:**
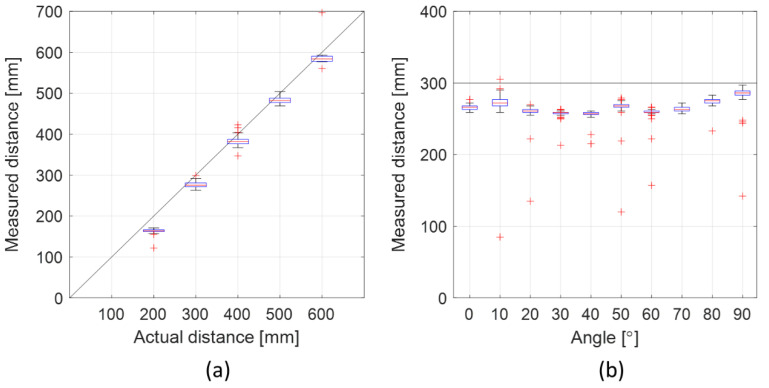
Lobe length and width characterization presented as boxplots (the red central line indicates the median, the bottom and top blue edges indicate the 25th and 75th percentiles, respectively, the black lines indicate the range of data points not considered outliers, and the red crosses indicate outliers). (**a**) Measured distance between the loudspeaker and the microphone compared to the actual distance. (**b**) Measured distance between the loudspeaker and the microphone for different loudspeaker angles at 30 cm distance. The solid black lines in (**a**,**b**) represent the points of equal actual and measured distances.

**Figure 7 sensors-21-02259-f007:**
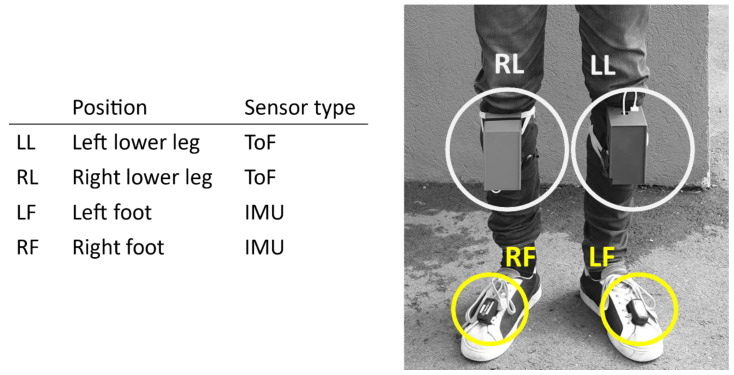
Sensor setup during dynamic gait test. The custom-made prototypes with ToF sensors are marked in light grey, and the Gait Up © IMUs are marked in yellow.

**Figure 8 sensors-21-02259-f008:**
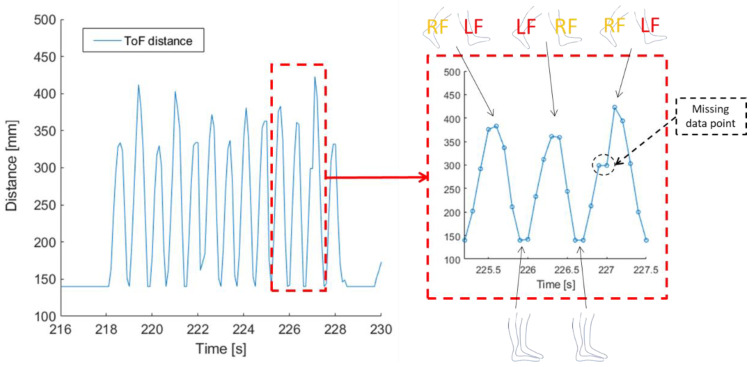
The semicontinuous ToF distance between the lower legs during a 25-foot walk. One full step during the walk is enlarged within the red dashed box visualizing the individual ToF distance measurements marked as blue circles. The relative positions of the feet are indicated above and below the plot for the extreme values of the ToF distance. The dashed circle indicates a missing ToF distance measurement (the value of the previous ToF distance measurement is given instead).

**Figure 9 sensors-21-02259-f009:**
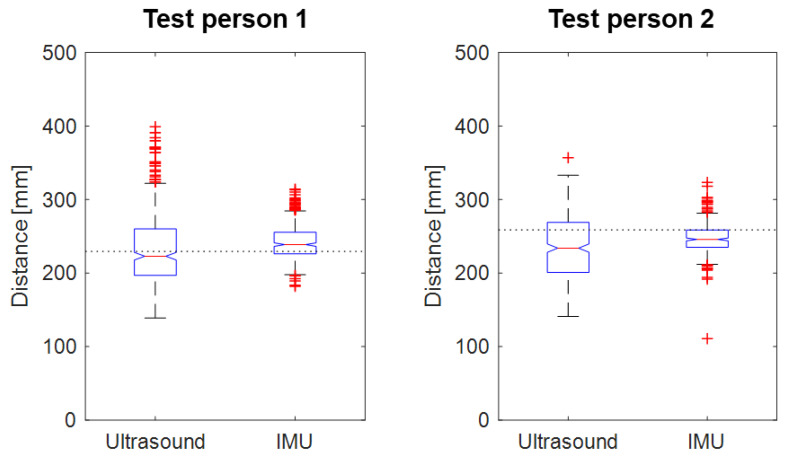
The maximum ToF distance (ultrasound) and estimated GaitUp ^®^ IMU stride length between the lower legs during 5-min treadmill walking by the two test persons presented as boxplots (the red central line indicates the median, the bottom and top blue edges indicate the 25th and 75th percentiles, respectively, the black lines indicate the range of data points not considered outliers, and the red crosses indicate outliers). The GaitUp ^®^ IMU stride length is pooled for both legs. The dotted line shows the average stride length between the lower legs as total walked distance divided by number of steps.

**Figure 10 sensors-21-02259-f010:**
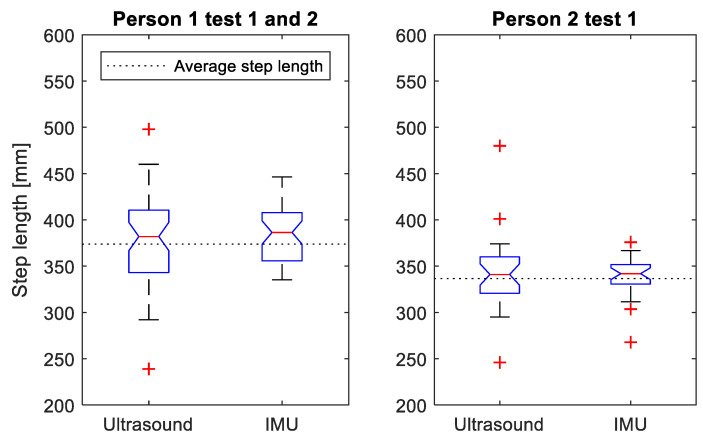
The maximum ToF distance and estimated GaitUp © IMU stride length between the lower legs during 25-foot walks for the two test subjects outdoors (two double tests by subject 1, and one double test by subject 2), presented as boxplots (the red central line indicates the median, the bottom and top blue edges indicate the 25th and 75th percentiles, respectively, the black lines indicate the range of data points not considered outliers, and the red crosses indicate outliers). The GaitUp ^®^ IMU stride length is pooled for both legs. The dotted line shows the average stride length between the lower legs as total walked distance divided by number of steps.

## Data Availability

The data presented in this study are available on request from the corresponding author.

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
