# Peer review of "Towards Human Motion Tracking Enhanced by Semi-Continuous Ultrasonic Time-of-Flight Measurements"

_sensors, 2021, doi:10.3390/s21072259_

Round 1

Reviewer 1 Report

General comments

the title should be modified since its focus is preliminary study using Human motion tracking enhanced by semi-continuous ultrasound time-of-flight measurements; I might argue their usefulness in terms of the study designthe same comment for data analysis methods
the study involve human participants; no reference to inclusion/non inclusion criteria, no reference to IWC and good medical practices, no clear definition of groups and variables under study no clear objective for this study; besides producing a instrumented by semi-continuous ultrasound it must be clear "what for "

Specific Comments

we should all be aware that experiments involving human participants or even data are currently under the scope of a very strict EU legislation applicable to all members states; the present study does not respect the basic requirements of that legislation ; related, the experimental design is poor, and inadequate to the general purpose of the study

A "Human motion tracking enhanced by semi-continuous ultrasound time-of-flight measurements " might be interesting, even for "normal" people that happen to have to improve physical function... it must be clear what kind of condition / stage, you willing to study and with what purpose since there is no evidence that  serve as a pathway human motion tracking is a valuable tool in several applications, including movement affected health condition assessments, in technique and performance analysis in sports, and in work life ergonomics evaluations. 

Reviewer 2 Report

  1. The aim of this article is to combine high frequency ultrasound sensors and IMU for gait spatial parameters estimation. It is hypothesized that the intersensory distance obtained from ultrasound sensors will improve the accuracy of spatial parameters and joint angles computation and yield to a system that is easier to configure and use.
  2. This topic is interesting, and the introduction clearly presents the justification of this work
  3. The main criticism that can be made of this article is the fact that, in the end, what is put forward in the introduction does not entirely correspond to the contribution of the article. It is stated that the combination of the ultrasound sensor and the IMU makes it possible to improve the spatial and angle data. The ultrasound sensors are tested on their own and it is not clear how the pooling of all sensor information (ultra sound and IMU) can be exploited. I think the contribution should be presented more clearly (Line100 -104 and lines 378-379).
  4. “Human motion tracking technology needs”, this paragraph is very informative, concise and sums up thoroughly the design and user needs.
  5. Line 177 : “signal had a signal to noise ratio adequate for this application” could you precise the value?
  6. Lien 214;” distance measurement is predicated” predicted
  7. Figure 6 left panel the measured distance and the actual distance differ by a factor 10, are the units correctly set? The vertical scale of the right panel graph could be reduced to have more details. What is the rms error of distance evaluation? What are the characteristics of the box plot (mean, median, sd interquartile?).
  8. Did you consider placing the sensors on the foot such as the GaitUp IMUs?
  9. Lines 308, 320, 379, 447, 449 “example of the measured semi-dynamic ToF distance” what do you mean by semi dynamic? Same for dynamic and non-dynamic.
  10. Line 341 : “The number of strides were found to correspond well between the different sensor measurements and the video records for both test persons.”. Could you please be more specific.

Reviewer 4 Report

Authors provided human tracking using ultrasound techniques which are very interesting ultrasound approach. In the human gait measurements, ToF distance data could be more accurate than typical ultrasound methods. As far as I know, time-of-flight measurement needs to be tested with error rates. The ultrasound setup for ToF measurement is correct. Authors showed some variances with some cases. However, there are only 2 persons case which are very small cases so that authors need to perform the proposed tests with at least 5 persons for scientific papers. If the measurement cannot be possible due to covid-19 situation, it is fine for this requirements. In addition, authors need to use professional English service or ask native English colleagues to improve the manuscript one more time because there are some broken English expression. In addition, authors missed some references and need to improve the Figures quality. Therefore, the manuscript can be improved if authors answer the following comments.

1. Figure 2 labels are small to be shown.
2. Figure 4 fonts are small to be shown.
3. Authors need to add the reference (Timing uncertainties can arise from delays in the protocol stack, both~) with the reference (Lazik, Patrick, et al. "Ultrasonic time synchronization and ranging on smartphones." 21st IEEE Real-Time and Embedded Technology and Applications Symposium. IEEE, 2015.) or another reference.
4. 1cm -> 1 cm in Line 215.
5. 30μs -> 30 μs in Line 216.
6. Authors need to add the reference (The transmitter and receiver boards are identical, except that the transmitter board is equipped with a loudspeaker while the receiver board is fitted with a microphone) with the reference (https://www.hindawi.com/journals/jhe/2017/6580217/)
7. Authors need to add the reference (To minimize the uncertainty, a simple real-time protocol (ESB) was chosen where both LBT (listen-before-talk) and retransmissions of lost packets may be switched off, as both these mechanisms introduce uncontrolled delays.) with the reference (Sutton, Gordon J., et al. "Enabling technologies for ultra-reliable and low latency communications: From PHY and MAC layer perspectives." IEEE Communications Surveys & Tutorials 21.3 (2019): 2488-2524.) or another related reference.
8. 100kHz -> 100 kHz in Line 222.
9.  Authors need to add the reference (The maximum sampling frequency provided by the nRF52 is 200kHz, which is somewhat low given~).
10. Authors need to add the reference (The algorithm employed for the energy detection can be expressed by the~)
11. Please correct (This can to a certain extent).
12. Please check MDPI reference styles because there are some missing information in conference papers.

Round 2

Reviewer 1 Report

The paper has much improved, and although I have reservations about the interpretation of the data, and the strength of evidence for the clinical message, I think the article presents the data well enough for readers to judge themselves. I would recommend publication.

Author Response

Thank you for recommend publication.

Reviewer 2 Report

The authors improved significantly the paper and responded to the vast majority of my comments. The aim of the paper has been reworked and corresponds to the contribution. The term semi dynamic is still unclear from a mechanical point of view and should be modified.

Author Response

Response to Reviewer 2 Comments

Point 1: The term semi dynamic is still unclear from a mechanical point of view and should be modified.

Response 1: Thank you for pointing this out, what we mean to say is "semi-continuous measurement of dynamic movement". We have updated this throughout the document. (in line 100, 278, 281, 338, 353, 418, 481, 483, 484 and 514)

Reviewer 3 Report

The authors addressed some of the issues only.

Line 100 "to investigate the feasibility of integrating a ToF sensor with IMUs"
I believe that this cannot be inserted in the "aim of the paper" because no step of sensor (IMU/ToF) fusion has been presented  neither discussed in the paper but this is only point (a hope) of a possible future task.
So, I think that line 103-107 could be a valid issue for the conclusion section not for the introduction one.

Figure 7 still report LL and RL --> sensor type: ToF + IMU
            The measure of IMU are not presented in the paper. I think that the figure should be modified

Line 414-416 In my opinion, it not possible to state that you study the "feasibility of building a sensor..." because this represents only a  your expectation. Indeed, no sensor fusion methodology neither any experimental proof has been given in the paper.

Author Response

Response to Reviewer 3 Comments

Point 1: Line 100 "to investigate the feasibility of integrating a ToF sensor with IMUs"

I believe that this cannot be inserted in the "aim of the paper" because no step of sensor (IMU/ToF) fusion has been presented neither discussed in the paper but this is only point (a hope) of a possible future task. So, I think that line 103-107 could be a valid issue for the conclusion section not for the introduction one.

Response 1: We have modified the aim of the study (line 99 and onwards) to read "The aim of the current study is to investigate the feasibility of using a high frequency (70-110 kHz) ultrasonic ToF sensor to enable dynamic measurements of relative limb-to-limb distance, and to build a prototype including both an IMU and a ToF sensor. In this study, we focus on testing a ToF sensor in dynamic gait tests and compare it with a commercial IMU. This is part of a larger effort to investigate if the relative dynamic distance between the IMUs can be used to improve the relative position estimation and angle estimation compared to using IMUs alone."

Hence, the aim is to build a prototype with both sensor types, and not to fuse the data.

We have kept the sentence starting at line 102 reading "This is part of a larger effort to investigate if the relative dynamic distance between the IMUs can be used to improve the relative position estimation and angle estimation compared to using IMUs alone." to give an indication on where this can be used in a larger context. However, we have removed the sentence "This might enable a system of IMUs that is easier to configure and use, and which yields more accurate and robust measurements of gait parameters in both in-lab and out-of-the-lab environments". The conclusion is already covering these topics, hence we have not added anything to the conclusion.

 Point 2: Figure 7 still report LL and RL --> sensor type: ToF + IMU

The measure of IMU are not presented in the paper. I think that the figure should be modified

Response 2: Figure 7 has been modified so that it does not to describe the integrated IMUs in the sensor prototype.

Point 3: Line 414-416 In my opinion, it not possible to state that you study the "feasibility of building a sensor..." because this represents only a your expectation. Indeed, no sensor fusion methodology neither any experimental proof has been given in the paper.

Response 3: Line 412 and onwards is modified to read: "We believe that ToF measurements of relative distance between the IMUs can be used to improve the relative position estimation and angle estimation compared to using IMUs alone. If this can be achieved, it might enable a system of IMUs that is easier to configure and use, and that yields more accurate and robust measurements of gait parameters in both in-lab and out-of-the-lab environments. In this study, the system has been tested for human gait both in a controlled lab environment on a treadmill, as well as an outdoor walk test. The system operates as intended and enables semi-continuous measurement of relative distances between the sensor modules, and with step length estimates comparable to commercial IMUs."

Reviewer 4 Report

Authors answered all the questions so the paper is accepted.

Author Response

Thank you for recommend publication.